# Impact of Pregnancy Rhinitis on Olfactory Sensitivity: A Controlled Comparative Study

**DOI:** 10.3390/diagnostics15202572

**Published:** 2025-10-13

**Authors:** Krystyna Sobczyk, Alicja Grajczyk, Karolina Dżaman, Justyna Zarzecka, Ewa Barcz

**Affiliations:** 1Department of Otolaryngology, Centre of Postgraduate Medical Education, Marymoncka 99/103, 01-813 Warsaw, Poland; krsobczyk@gmail.com; 2Department of Otolaryngology, Collegium Medicum, Faculty of Medicine, Cardinal Stefan Wyszynski University, Miedzylesie Specialist Hospital, 01-938 Warsaw, Poland; alicja.grajczyk@gmail.com; 3Department of Gynecology and Obstetrics, Collegium Medicum, Faculty of Medicine, Cardinal Stefan Wyszynski University, Miedzylesie Specialist Hospital, 01-938 Warsaw, Poland; jmgilewska@gmail.com (J.Z.); e.barcz@uksw.edu.pl (E.B.)

**Keywords:** pregnancy, smell disturbance

## Abstract

**Background/Objectives**: During pregnancy, the body undergoes numerous physiological changes, many of which are driven by significant hormonal shifts. Pregnancy rhinitis is a condition characterized by nasal congestion that occurs during pregnancy without any other signs of respiratory infection or known allergic causes. The aim of the study was to examine the impact of pregnancy rhinitis on the sense of smell. Specifically, it focused on determining how the nasal congestion associated with pregnancy rhinitis may alter olfactory perception in pregnant people. **Methods**: The study group comprised fifty women, aged 18 to 41, all in their third trimester of pregnancy. The control group was made up of 25 non-pregnant women between the ages of 25 and 31. Olfactory function was assessed using Sniffin’ Sticks, and each participant completed the SNOT-22 questionnaire. Additionally, ENT examination, nasofiberoscopy, rhinomanometry were performed. **Results**: The comparison between the control and study groups in terms of detection, discrimination, and identification test scores revealed statistically significant differences. The study group demonstrated lower odor average test scores, indicating worse olfactory acuity and poorer identification abilities, with these effects being strong. In addition, the study group showed a lower discrimination test score compared to the study group, though this effect was weak. On the other hand, the control group showed a higher level of discrimination test score compared to the study group, though this effect was weak. However, the pregnant women did not perceive any subjective impairment in their sense of smell even though they had smell disturbances confirmed in the Sniffin Stick test. The SNOT-22 questionnaire results indicated that the study group reported subjectively worse nasal patency compared to the control group. **Conclusions**: This controlled study demonstrated that olfactory disturbances, confirmed by the Sniffin’ Sticks test, affected half of the pregnant participants, with reduced smell sensitivity observed in advanced pregnancy compared to non-pregnant controls. Notably, more than half of the women with objectively confirmed olfactory deficits did not report subjective complaints, highlighting the need for greater clinical awareness of sensory changes during pregnancy. Pregnancy-related swelling of the nasal mucosa leads to impaired upper airway airflow, contributing to a reduction in olfactory sensitivity.

## 1. Introduction

Pregnancy represents a physiological state characterized by extensive systemic alterations driven by significant hormonal fluctuations. The hormonal changes, particularly elevated levels of estrogen and progesterone, are crucial for maintaining pregnancy but also exert widespread effects on multiple organ systems, including the respiratory and olfactory systems. One common outcome of these changes is pregnancy rhinitis [1], a condition marked by persistent nasal congestion during pregnancy in the absence of respiratory infection or allergic triggers. Pregnancy rhinitis typically emerges during the second or third trimester and affects a substantial proportion of pregnant women [2]. Although estimates of its prevalence vary, research indicates that up to 30% of pregnant people may experience this condition [3]. Despite its frequent occurrence, pregnancy rhinitis remains underdiagnosed and undertreated, with its symptoms often misattributed to other causes or dismissed as a normal aspect of pregnancy [4]. This condition can significantly affect respiratory function and, in turn, may influence olfactory perception, leading to altered sensory experiences [5,6].

The sense of smell is a critical sensory function closely tied to the respiratory system, as olfactory receptors are located within the nasal passages. Any disruption in nasal airflow—such as that caused by pregnancy-induced rhinitis (PIR)—can impair olfactory sensitivity. Pregnant women frequently report changes in olfactory perception, ranging from heightened sensitivity to certain odors to a generalized reduction in olfactory acuity. These sensory alterations may contribute to various pregnancy-related symptoms, including food aversions, nausea, and amplified emotional responses, ultimately affecting overall quality of life [3,4,5,7].

Despite the recognized prevalence of PIR and its potential effects on the sense of smell, research exploring the direct relationship between PIR and olfactory function remains limited. For ENT (Ear, Nose, and Throat) specialists, understanding this relationship is critical for several reasons. First, PIR can substantially affect a pregnant woman’s respiratory function and overall comfort, making it a relevant concern for ENT practitioners [8,9]. Addressing nasal congestion and associated symptoms, such as diminished olfactory function, can enhance quality of life during pregnancy, especially during the third trimester when breathing difficulties may exacerbate other discomforts, including fatigue and sleep disturbances [10]. Moreover, olfactory impairments associated with PIR may influence various pregnancy-related factors, such as nutrition and psychological well-being. A diminished sense of smell may reduce appetite, exacerbate food aversions, and potentially lead to weight loss, all of which can have significant implications for both maternal and fetal health. Additionally, since olfaction is closely linked to emotional processing, a reduced sense of smell may contribute to heightened stress or mood disturbances, further complicating the emotional challenges that many women encounter during pregnancy [11].

For ENT professionals, comprehensively understanding the impact of PIR on olfactory function extends beyond managing nasal congestion. It involves recognizing the broader health implications for pregnant people, particularly concerning their respiratory, nutritional, and psychological well-being, thus enabling more effective and holistic care. Therefore, the study aimed to examine the impact of pregnancy rhinitis on the subjective and objective sense of smell. Specifically, it focused on determining how the nasal congestion associated with pregnancy rhinitis may alter olfactory perception in pregnant people [4,12].

## 2. Materials and Methods

The study involved a group of fifty women between the ages of 18 and 41, all in their third trimester of pregnancy (between the 28th and 41st week). These patients were examined at the Department of Otolaryngology at Cardinal Stefan Wyszyński University, Międzylesie Specialist Hospital in Warsaw, and later admitted to the Department of Gynecology and Obstetrics at the same institution. The control group consisted of 25 healthy non-pregnant women, primarily medical students, aged between 24 and 32 years (mean age 26 ± 2.58).

Participants with a history of smoking, anatomical abnormalities, or conditions that could influence nasal airflow were excluded from the study. Each participant underwent a comprehensive ENT examination, including anterior and posterior rhinoscopy and fiberoscopic assessment of the nasal cavities. The study was designed as a cross-sectional analysis focusing exclusively on the third trimester of pregnancy. This approach was chosen because symptoms of pregnancy rhinitis and associated hormonal and vascular changes in the nasal mucosa are most pronounced during this period, increasing the likelihood of detecting clinically relevant nasal obstruction and olfactory alterations.

To evaluate changes in the nasal cavity patency, rhinomanometry was conducted using the “COMBI 4000” ENT diagnostic module (Homoth Medizinelektronik GmbH & Co KG Kaltenkirchen, Germany). Rhinomanometry was performed by inserting nasal probes into the patient’s nostrils while they breathed naturally. The breathing intensity was monitored in real time on the screen. The device measured airflow in mL/s for both inhalation and exhalation at pressures of 75, 150, and 300 Pa, separately for the left and right nostrils. Additionally, it calculated total airflow and the corresponding resistance coefficients.

Olfactory function was evaluated with the Sniffin’ Sticks test [13], which consists of three subtests: odor threshold test (OTT), odor discrimination test (ODT), and odor identification test (OIT). In the OTT (odor threshold test), participants were presented with felt—tip pens containing serial dilutions of n-butanol, starting from the lowest concentration. Using a three-alternative forced-choice procedure, in each trial participants were asked to identify which of three pens contained the odorant. The concentration was increased stepwise until the odorant was correctly detected in two consecutive trials, and then decreased again following a staircase paradigm to determine the lowest reliably detected concentration. The final threshold score represented the mean of the last four staircase reversals. Higher scores indicated better olfactory sensitivity (i.e., detection of lower odorant concentrations), whereas lower scores reflected reduced sensitivity. In the ODT, participants were presented with sets of three pens, two with the same odor and one with a different one, to assess their ability to differentiate between scents. The third test—OIT used felt-tip pens (“sticks”) filled with different odorants. During the test the subject was presented with 16 different odor sticks, one at a time. For each stick, the patient chose the correct smell from four multiple-choice options. Total scores from all these tests were compared to standardized norms to assess overall olfactory function. Based on the results, participants were diagnosed with normosmia, hyposmia, or anosmia, as determined by placing their overall Sniffin’ Sticks score on a centile chart.

In addition, participants completed the SNOT-22 (Sino-Nasal Outcome Test) questionnaire, which is widely used in otolaryngology to assess the severity of symptoms related to chronic rhinosinusitis and other sinonasal conditions. This 22-item questionnaire addresses symptoms like nasal blockage, discharge, facial pain, and olfactory impairment, as well as the impact of these symptoms on daily life. While the SNOT-22 is valuable, it has limitations due to its reliance on self-reported data, which may be subject to recall bias. In the context of pregnancy-related studies, like the one described, the SNOT-22 helps to evaluate how pregnancy-induced rhinitis (PIR) and related nasal symptoms affect not only physical health but also emotional well-being. PIR can lead to significant nasal blockage, impacting olfactory function and overall quality of life, which this tool can measure comprehensively. By using SNOT-22, researchers can quantify the burden of PIR and compare it with control groups, providing insights into how sinonasal issues may differ in pregnant people versus the general population.

Statistical analyses were conducted using IBM SPSS Statistics version 29. Descriptive statistics (mean, median, standard deviation, and range) were calculated for all variables. Distribution normality was tested using the Shapiro–Wilk test. Between-group comparisons were performed using the Mann–Whitney U test for continuous variables and Fisher’s exact test for categorical data. Effect sizes were calculated (η^2^). Correlation analyses between subjective olfactory measures (SNOT-22) and objective olfactory test results (Sniffin’ Sticks) were performed using Pearson’s correlation when normality assumptions were met and Spearman’s rank correlation when they were not. All *p*-values are reported as exact values, with α = 0.05 considered the threshold for statistical significance.

The study received ethical approval from the Bioethics Committee of the Regional Medical Chamber in Warsaw, with registration number KB/1440/23.

## 3. Results

### 3.1. Olfactory Results

#### 3.1.1. Analysis of Self-Reported Subjective Olfactory Problems (SNOT-22)

All participants responded to a specific question from the SNOT-22 questionnaire regarding subjective olfactory disorders. Responses were recorded on a Likert scale ranging from 0 to 5, where higher scores indicated greater perceived impairment.

The results showed no statistically significant difference between the study and control groups in terms of subjective olfactory dysfunction (*p* = 0.261, η^2^ = 0.558), suggesting that both groups perceived their sense of smell similarly.

#### 3.1.2. Sniffin Sticks Test Results in Terms of Detection, Discrimination, and Identification Test Scores

Basic descriptive statistics and group comparisons were conducted for detection test score (OTT), discrimination test score (ODT), and identification test score (OIT) (Table 1). The study group demonstrated significantly lower scores in all three olfactory function domains compared to the control group. The Mann–Whitney U test revealed statistically significant differences in odor test score (Z = −4.63, *p* < 0.001, η^2^ = 0.290) and identification test score (Z = −4.14, *p* < 0.001, η^2^ = 0.230), indicating moderate to large effect sizes.

A significant difference was also observed in the discrimination test score (Z = −2.01, *p* = 0.044, η^2^ = 0.050), though with a smaller effect size. These findings suggest that the study group experienced a notable decline in olfactory performance, particularly in odor detection and identification, while the effect on discrimination ability was less pronounced.

#### 3.1.3. Olfactory Diagnosis Based on Sniffin Sticks Test Results

Based on the total summary result from OIT, ODT, and OTT scores, the total Sniffin Stick test result was calculated and compared to standardized norms to assess overall olfactory function. Participants were diagnosed with normosmia, hyposmia, or anosmia, as determined by placing their overall Sniffin’ Sticks score on a centile chart, as follow: supersmeller (≥41.50), normosmia (41.25–30.75), hyposmia (30.50–16.25), and anosmia (≤16.00). Each participant was assigned a numerical value corresponding to their diagnostic category (1 = Supersmeller, 2 = Normosmia, 3 = Hyposmia, 4 = Anosmia), which was then used for statistical analysis (Table 2).

In the study group, 54% (27/50) of pregnant women were found to have olfactory impairments, with one individual experiencing complete anosmia. In contrast, only 4% (1/25) of participants in the control group exhibited impaired olfaction, all of whom had hyposmia.

Statistical analysis revealed a significantly higher prevalence of olfactory dysfunction in the study group compared to the control group, indicating a notable difference in olfactory function between the two populations.

#### 3.1.4. Analysis of the Relationship Between the Subjective Assessment of Olfactory Sense and the Results of Sniffin Stick Test

In addition to the Sniffin’ Sticks test, self-reported olfactory dysfunction was evaluated using the SNOT-22 questionnaire, which measures subjective nasal and olfactory symptoms. The comparison between objective olfactory function (Sniffin’ Sticks) and subjective olfactory perception (SNOT-22) provided a comprehensive analysis of olfactory impairment in the study group.

Correlation analysis revealed a moderate positive association between the Sniffin’ Sticks diagnosis and the SNOT-22 olfactory dysfunction scores. Pearson’s correlation indicated r = 0.38, *p* = 0.030. Given the borderline normality of the distribution, we also conducted Spearman’s rank correlation, which yielded rho = 0.36, *p* = 0.047. Both tests consistently demonstrated a moderate positive association. Importantly, all *p*-values are reported as exact values, and *p* = 0.050 was not considered statistically significant.

These findings suggest that although objective testing frequently identified hyposmia or anosmia, many patients did not subjectively perceive olfactory impairment. In other words, despite measurable deficits in the Sniffin’ Sticks test, patients often reported relatively low SNOT-22 scores regarding smell, indicating a potential underestimation of their dysfunction.

### 3.2. Nasal Patency Results

#### 3.2.1. Analysis of Self-Reported Subjective Nasal Obstruction (SNOT-22)

A comparative analysis was conducted to evaluate the reported frequency of subjective nasal obstruction between the control and study groups. In the control group, 40% of participants (10/25) reported nasal obstruction, compared with 96% (48/50) in the study group. This difference was assessed using Fisher’s exact test (*p* = 1.38 × 10^−7^; OR = 0.028), confirming a significantly higher prevalence of nasal obstruction among pregnant women. Additionally, group differences in mean obstruction scores (Likert 0–5 scale) were analyzed using the Mann–Whitney U test (Z = −7.85, *p* < 0.001), which further supported the significant subjective impairment in the study group. All participants answered a specific item from the SNOT-22 questionnaire assessing subjective nasal obstruction, rated on a Likert scale from 0 to 5, with higher scores indicating greater perceived nasal blockage.

The analysis demonstrated a statistically significant difference between the study and control groups, with the study group reporting markedly higher subjective nasal obstruction scores (*p* < 0.001, η^2^ = 0.937). These findings indicate that participants in the study group experienced substantially greater perceived impairment in nasal airflow compared to those in the control group.

#### 3.2.2. Rhinomanometry Results

Rhinomanometry revealed marked differences between the study and control groups. In the study group, nasal airflow values ranged from 0 to 487 mL/s. Within-group analysis demonstrated that airflow decreased significantly as measurement pressure increased (Wilcoxon signed-rank test, *p* < 0.001 for all parameters), confirming a progressive increase in nasal resistance among pregnant women.

In the control group, airflow ranged from 98 to 436 mL/s, and repeated measurements showed that the applied pressure levels did not significantly influence nasal airflow (Wilcoxon signed-rank test, *p* > 0.05). This indicates that healthy controls maintained stable nasal patency across different pressure levels.

For between-group comparisons at each pressure level, the Mann–Whitney U test was performed and consistently demonstrated significantly reduced airflow in the study group compared with controls (all *p* < 0.001). To better illustrate the magnitude of these differences, we calculated the mean difference (MD = 2.04) with a 95% confidence interval [1.64–2.44], *p* < 0.001, indicating a robust and clinically meaningful impairment of nasal patency in pregnant women.

#### 3.2.3. Analysis of the Relationship Between the Subjective Assessment of Nasal Patency and the Results of Rhinomanometry

The relationship between subjective (nasal obstruction in SNOT-22) (Figure 1) and objective (rhinomanometry results) assessments of nasal patency in the study group was examined using Spearman’s correlation, given the nonparametric nature of the variables. The results demonstrated negative correlations between all rhinomanometry parameters and subjective nasal obstruction (rho ranging from −0.69 to −0.62, *p* < 0.001 for all parameters) (Table 3). Higher self-reported nasal obstruction was associated with lower airflow as measured by rhinomanometry, with the strength of these relationships classified as moderate to strong.

### 3.3. The Relationship Between Assessment of Nasal Patency and Objective Examination of the Sense of Smell in Pregnant Women

In pregnant women, the mean olfactory threshold score was significantly lower compared with the control group (7.06 ± 2.67 vs. 10.36 ± 1.28; Z = −4.63, *p* < 0.001, η^2^ = 0.29). Similarly, odor identification was reduced in the pregnant cohort (11.82 ± 1.88 vs. 13.56 ± 1.04; Z = −4.14, *p* < 0.001, η^2^ = 0.23), while odor discrimination showed a smaller but still significant difference (11.36 ± 1.57 vs. 12.04 ± 0.84; Z = −2.01, *p* = 0.044, η^2^ = 0.05). To further examine whether these differences varied within pregnancy, we compared olfactory scores across trimesters. This analysis was conducted exclusively in the pregnant cohort, as the control group represented a single homogeneous sample of non-pregnant women and therefore could not be stratified by trimester.

## 4. Discussion

The present study demonstrated that pregnant women with pregnancy-induced rhinitis (PIR) exhibited significantly associated olfactory function compared with healthy non-pregnant controls. Specifically, olfactory threshold, discrimination, and identification scores were all impaired, with the most pronounced difference observed for odor threshold. These findings indicate that PIR may contribute to both subjective and objective olfactory dysfunction during pregnancy [14,15]. Our results showed a significant association between the degree of nasal obstruction [10] and rhinomanometric airflow parameters, suggesting that mechanical airway limitation is an important contributor to impaired olfactory performance in this population [12,13,14].

Importantly, this study revealed a notable discrepancy between subjective and objective assessments of olfactory function. While many pregnant women did not report olfactory disturbances in the SNOT-22 questionnaire, the Sniffin’ Sticks test indicated measurable impairment. This mismatch is clinically relevant, as unrecognized olfactory dysfunction may affect daily life, including food choices, safety, and emotional well-being. Notably, such discrepancies are not unique to pregnancy, as previous research has consistently shown that self-reported smell function is only weakly correlated with psychophysical test results in the general population [16]. In contrast, we observed that self-reported nasal obstruction did correlate with rhinomanometric airflow parameters, underscoring the validity of subjective reporting for nasal patency but not for olfactory capacity [17,18,19,20].

The relationship between olfaction and pregnancy-related conditions such as hyperemesis gravidarum has also been explored in prior literature [21,22]. Some studies propose that altered olfactory perception contributes to nausea and vomiting during pregnancy, although findings are inconsistent. The present study was not designed to evaluate this association directly, but the potential link between olfactory dysfunction and gastrointestinal symptoms warrants further investigation.

When compared with earlier studies on olfactory function during pregnancy, our results align with those showing reduced sensitivity in late pregnancy. However, previous research has produced mixed findings, with some reporting heightened sensitivity and others noting no significant changes [23,24,25]. These discrepancies are likely influenced by methodological differences, sample sizes, and the gestational periods studied. This underscores the need for standardized olfactory testing protocols and longitudinal research covering all trimesters.

The observed reduction in odor threshold and identification scores in the pregnant group suggests that pregnancy-related changes may impair olfactory sensitivity. Mechanical obstruction of nasal airflow may impede the delivery of odorant molecules to the olfactory epithelium, thereby impairing the sense of smell. Similar relationships between nasal congestion and olfactory deficits have been reported in studies on both allergic and non-allergic rhinitis [15,16,17].

In addition to mechanical factors, hormonal influences are likely to play a role in pregnancy-related olfactory changes [17]. Experimental and clinical research has suggested that hormonal fluctuations—particularly in estrogen, progesterone, oxytocin, and prolactin—may modulate olfactory sensitivity [19,24,25,26]. These findings are supported by animal studies, which have demonstrated that exogenous administration of these hormones can directly affect olfactory processing. While these results are compelling, further research is needed to confirm their relevance in human pregnancy.

It is worth noting that while low SNOT-22 scores generally correspond to lower objective olfactory performance, some participants may report high SNOT-22 scores and still achieve high Sniffin’ Sticks results. This could be due to factors such as nasal obstruction, sleep disturbances, or general discomfort affecting their self-reported sinonasal burden, even when their objective olfactory function remains intact. This highlights the importance of considering both subjective and objective measures when evaluating olfactory performance.

Despite its strengths, this study has several limitations. First, it focused exclusively on the third trimester; future studies should assess nasal and olfactory function across all stages of pregnancy to better understand temporal patterns. While the third trimester is recognized as the peak phase for pregnancy rhinitis symptoms, a longitudinal assessment across all trimesters would provide a more comprehensive understanding of the trajectory of olfactory changes during pregnancy. Second, although the SNOT-22 questionnaire provided insight into subjective symptoms, it is not specifically validated for olfactory assessment. The inclusion of olfaction-specific questionnaires or additional psychophysical testing may enhance future investigations [26,27].

Another limitation of this study is the difference in age distribution between the study and control groups. The control group consisted of healthy medical students, whereas the study group comprised pregnant women attending prenatal care. Although olfactory function may decline slightly with age, all participants in our study were under 42 years of age, and previous normative data obtained with the Sniffin’ Sticks test indicate that this age range is unlikely to introduce substantial olfactory bias. Nevertheless, we acknowledge this as a potential confounding factor and have addressed it accordingly. Moreover, smoking is a recognized factor influencing olfactory performance; therefore, in our study, only non-smokers and former smokers were included. While this criterion minimized the confounding effect of active smoking, other potential co-factors—such as environmental exposures or mild upper respiratory tract infections—could not be fully controlled and may have influenced olfactory outcomes.

## 5. Conclusions

This study demonstrated that pregnant women with pregnancy-induced rhinitis exhibit significantly reduced olfactory function compared with healthy non-pregnant controls. The most pronounced difference was observed in odor threshold scores, accompanied by impairments in odor discrimination and identification. These findings indicate that both subjective and objective olfactory dysfunction may occur in pregnancy and that nasal obstruction contributes substantially to this impairment. Importantly, while self-reported nasal obstruction correlated with rhinomanometric measurements, self-reported olfactory performance did not align with psychophysical test results, underscoring the need for objective assessment. Taken together, these results highlight the clinical relevance of pregnancy-induced rhinitis for ENT practice and suggest that systematic evaluation of nasal patency and olfactory function may improve the quality of care for pregnant women.

Pregnancy-related swelling of the nasal mucosa may lead to impaired upper airway airflow, contributing to a notable reduction in olfactory threshold results. These changes can significantly impact the quality of life of pregnant patients.

## Figures and Tables

**Figure 1 diagnostics-15-02572-f001:**
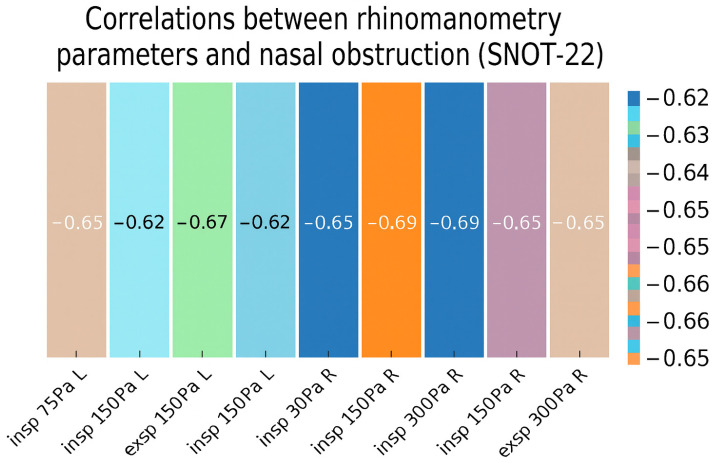
Heatmap of correlations between rhinometry parameters and nasal obstruction.

**Table 1 diagnostics-15-02572-t001:** Results of Sniffin Stick test in the study and control group.

Dependent Variable	Mean (Control)	SD (Control)	Mean (Study)	SD (Study)	z	*p*	η^2^
Olfactory Threshold Test Score (OTT)	10.36	1.28	7.06	2.67	−4.63	<0.001	0.290
Discrimination Test Score (ODT)	12.04	0.84	11.36	1.57	−2.01	0.044	0.050
Identification Test Score (OIT)	13.56	1.04	11.82	1.88	−4.14	<0.001	0.230

**Table 2 diagnostics-15-02572-t002:** Olfactory diagnosis based on Sniffin Stick test results.

Diagnosis	Study Group	Control Group
Normosmia	23	24
Hyposmia	26	1
Anosmia	1	0
Total	50	25

**Table 3 diagnostics-15-02572-t003:** Comparison of SNOT-22 responses on nasal obstruction and olfactory disturbances between control and study groups.

Dependent Variable	M (Control)	Me (Control)	SD (Control)	M (Study)	Me (Study)	SD (Study)	Z	*p*	η^2^
Nasal obstruction	0.44	0.0	0.583	2.48	2.00	1.165	1171.5	<0.001	0.937
Olfactory disturbances	0.16	0.0	0.374	0.56	0.0	1.072	697.0	0.261	0.558

## Data Availability

The original contributions presented in this study are included in the article. Further inquiries can be directed to the corresponding author.

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
