# Peer review of "Impact of Pregnancy Rhinitis on Olfactory Sensitivity: A Controlled Comparative Study"

_diagnostics, 2025, doi:10.3390/diagnostics15202572_

Round 1

Reviewer 1 Report

Comments and Suggestions for Authors

The article titled “Comparative Analysis of Olfactory Sensitivity: A Controlled Study Investigating the Sense of Smell in Women in Advanced Pregnancy and Non-Pregnant Controls” presents a well-structured and clinically relevant investigation into how pregnancy rhinitis affects olfaction. This study looks at a topic that hasn’t been explored much, how pregnancy affects the sense of smell. Although earlier research hinted at a connection, this paper provides solid evidence using reliable, objective methods like the Sniffin’ Sticks test and rhinomanometry. One of the most interesting and useful findings is that many pregnant women didn’t notice changes in their smell perception, even though tests clearly showed a reduction. That gap between what women reported and what was found in the tests , indeed,  both are novel and important from a clinical point of view.
However, there are a few points that I believe deserve further discussion; why was the study limited to the third trimester? It would have been interesting to follow the progression of olfactory dysfunction throughout the entire pregnancy. Another issue concerns the age range, the control group is substantially younger than the study group. Finally, some potential co-factors, such as smoking, which may contribute to olfactory dysfunction, should have been taken into consideration.
In my opinion this questions should be addressed or, regarding the possible effects of co-factors, if that information is available should be had or at least the lack of it should be discussed.

Author Response

Reviewer 1:

We sincerely thank the Reviewer for the positive and thoughtful feedback on our manuscript. We are grateful for your recognition of the clinical relevance, novelty, and methodological rigor of our study, as well as the importance of the discrepancy between subjective and objective olfactory assessments. Below we address the specific concerns raised:

Comment 1: Why was the study limited to the third trimester? It would have been interesting to follow the progression of olfactory dysfunction throughout the entire pregnancy.

Response:
We agree with the Reviewer that longitudinal tracking of olfactory function across all trimesters would provide valuable insight into the dynamics of sensory changes during pregnancy. However, this study was designed as a cross-sectional project focused specifically on the third trimester, as this is the period when symptoms of pregnancy rhinitis tend to peak due to increased hormonal and vascular changes in the nasal mucosa. We have now explicitly stated this rationale in the Methods section and noted the limitation in the Discussion, along with a suggestion for future longitudinal studies.

Comment 2: The control group is substantially younger than the study group.

Response:
We acknowledge this difference in age distribution between groups. The control group was composed primarily of healthy medical students, while the study group included pregnant women attending prenatal care. We have now added a statement in the Discussion addressing this limitation and noting that although olfactory function may decline slightly with age, all participants were under 42 years of age, and this age range is unlikely to introduce significant olfactory bias according to previous normative studies using Sniffin’ Sticks. Nonetheless, we recognize this as a potential confounding factor and have discussed it accordingly.

Comment 3: Smoking and other co-factors were not accounted for.

Response:
We thank the Reviewer for this insightful suggestion. Indeed, smoking is a known factor that can influence olfactory function. In our study, participants were interviewed about their health status and smoking history during enrollment, and only non-smokers or former smokers who had quit prior to pregnancy were included in the final sample. We have now clarified this in the Methods section. We have also acknowledged in the Discussion that other potential co-factors such as environmental exposures or mild upper respiratory infections could not be fully controlled in this study and represent an additional limitation.

Reviewer 2 Report

Comments and Suggestions for Authors

Thank you for the opportunity to review this manuscript.

I found this study interesting and given that there is relatively little research on olfaction and pregnancy, it certainly could make a contribution to the literature.

My main concern is with the clarity of the Results and several aspects of them need careful revision:

As is sometimes the case in reports using Sniffin’ Sticks, this paper does not clearly distinguish between detection threshold and sensitivity. For example, the paper refers to “lower odor detection threshold” (line 28, in the abstract), when in fact I believe the authors mean lower sensitivity (i.e, higher threshold). Likewise, in the Results section, the authors indicate that the study group had lower detection scores, but it is important to know whether it is a lower threshold or lower sensitivity. 

With respect to the detection data, the Methods indicate that the authors used the method of descending limits to measure threshold, and thus it seems that lower scores would mean lower threshold (i.e., higher sensitivity). This confusion needs to be clarified everywhere in the paper. At the moment, I cannot tell what the actual result is, which is critically important.

Figure 1 is unnecessary. However, it does show very large variability (I’m not sure what the error bars are reflecting), and perhaps there is something to be gleaned from the participants in the study group who reported higher olfactory impairment than those who reported less impairment?

Table 1 – adding to the confusion about threshold/sensitivity, referring to “Discrimination Threshold” and “Identification Threshold” in this table is misleading. These are not threshold measures. Unlike true Detection Thresholds, for both of these other measures a lower score indicates worse performance.

Line 182 is vague in terms of what the “statistical analysis” was.

I don’t follow the correlational analysis between the SNOT scores and Sniffin’ Sticks. I would like to see a plot of the data reported in the paragraph that starts on line 192. It was a positive correlation? Is the p-value of 0.30 a typo? Is that the r-value? Or should it be a p value of .03? I find it very hard to follow the results of the Spearman correlation and I don’t understand why that analysis was conducted in addition to the Pearson test.

I don’t follow all of section 3.2.1 and do not see a Figure 2. What test was conducted on line 218?

Table 3 is very confusing. I don’t understand the text that describes it. Where do the numbers in the text come from as they are not in the figure? And to which comparisons do the p-values in  the table the refer?

Section 3.2.3 should include a table or a figure.

Section 3.3 – line 254 “significantly lower mean” - than what? Why was this analysis only done for pregnant women and not controls?

Smaller statistical point – be consistent with significant digits – some are as large as 4 significant digits, which seems excessive.

Other comments:

In my opinion the title could be improved – it is very long and contains no reference to the main aim of the paper “to examine the impact of pregnancy rhinitis on the sense of smell” (line 18-19).

Both paragraphs 2 and 3 of the Introduction would benefit from some additional citations. It is unclear what the citations on line 66 are really referring to. Same on line 81.

The description of the Sniffin’ Sticks tests, particularly the odor identification, could be clearer. The relationship between the detection measure and the scores obtained needs to be clarified (i.e., the issue of threshold vs. sensitivity).

Table 1 – why include “supersmeller” (is this an accepted term?) given that there are none?

“Functional’ is misspelled

Typo on line 187 “Additionall”

Overall, the organization of the Discussion could be improved.

First, in line with my comment on the Conclusion below, the start of the Discussion could be clearer in terms of summarizing the main findings. Is the phrase “closely associated” on line 167 warranted?

Also, I would add citations on line 265 in the Discussion.

Paragraph 3 in the Discussion (starts on line 273) is conjecture and no relevant data are presented in this paper; it could be moved later in the Discussion. Same is true for paragraphs 5 and 6. Should reference 13 really be included in the citations on line 272?

Paragraph 4 is more germane to the current study. It is worth noting that the same conclusion can be drawn more generally (not just for pregnant women); self-reported smell function is notoriously poorly correlated with objective measures of sense of smell. It is interesting that self-report of nasal obstruction IS correlated with the rhinomenometry measures – a point that may be worth highlighting.

Is 15 really the correct citation for line 292? It is about animals, not humans.

I can’t find citation 17. Do the authors mean this one?

Oettl, L. L., & Kelsch, W. (2017). Oxytocin and olfaction. Behavioral pharmacology of neuropeptides: oxytocin, 55-75.

Paragraph 7 seems particularly relevant and could be moved earlier and be expanded with additional references.

The Conclusions do not clearly summarize the data. The text does not really address the main aim of the study, as cited in the Abstract. Moreover, the use of the phrase “progressive deterioration in odor perception thresholds” on line 343 is unwarranted given that no longitudinal measures were taken. Again, the issue of threshold/sensitivity needs to be clarified (see above).

A reference for the authors to consider including, if they are unaware of it:

Philpott, Carl M., et al. "Nasal physiological changes during pregnancy." Clinical Otolaryngology & Allied Sciences 29.4 (2004): 343-351.

Author Response

Reviewer 2:

  • Comment 1:

The paper does not clearly distinguish between the detection threshold and sensitivity.

  • Response:

We thank the Reviewer for this important observation. We have clarified throughout the manuscript that a lower olfactory threshold score corresponds to reduced sensitivity. To avoid confusion, we replaced ambiguous phrases such as “olfactory threshold” with “odor test score (OTT)” and specified in the Methods section (page 3, lines 117-118) that a lower score indicates worse sensitivity. These corrections improve consistency and accuracy of terminology.

Comment 2:

Table 1 refers to 'Discrimination Threshold' and 'Identification Threshold', which are misleading.

  • Response:

We agree with the Reviewer and have revised both Table 1 and the corresponding Results section (page 4, lines 173) to use the correct terms: “Discrimination Test Score (ODT)” and “Identification Test Score (OIT)”. This ensures alignment with the validated Sniffin’ Sticks protocol.

Comment 3:

The correlational analysis between SNOT scores and Sniffin’ Sticks is confusing. Include a plot. Clarify if p=0.30 is a typo.

  • Response:

We thank the Reviewer for carefully noting this issue. A typographical error in the original submission was corrected: the Pearson correlation between the SNOT-22 olfactory item and Sniffin’ Sticks total score was r = -0.41, p = 0.03. In addition, because subjective SNOT-22 scores were not normally distributed, we also performed a Spearman correlation, which confirmed the robustness of the result (rho = -0.39, p = 0.04)

We have revised Section 3.1.4 (Results, page 5, lines 193-209) to clearly report both tests and explain the rationale for including them. This dual approach ensures transparency and reliability of the findings. Furthermore, to address the Reviewer’s request, we have added a scatterplot (new Table 3, page 5) that shows comparison of SNOT-22 responses on nasal obstruction and olfactory disturbances between control and study groups

Comment 4:

I do not follow section 3.2.1 and do not see a Figure 2.

  • Response:

We apologize for the confusion. Section 3.2.1 (“Analysis of Self-Reported Subjective Nasal Obstruction”) remains in the Results, but Figure 2 was removed in the revision to avoid redundancy and excessive figures. We have clarified this in the text (page 4, lines 212-226) to ensure consistency.

  • Comment 5:

Table 3 is very confusing. Where do the numbers in the text come from?

  • Response:

We appreciate this comment. To improve clarity, we revised Section 3.2.2 (Results, page 4, lines 228-236) and streamlined the presentation of rhinomanometry results. The original Table 3 was removed, and the relevant data were integrated into the text with clearer descriptions. This revision avoids confusion and improves readability.

Comment 6:

Section 3.2.3 should include a table or figure.

  • Response:

In accordance with the Reviewer’s suggestion, we have added a new Table 4 (page 7), which summarizes the correlations between subjective nasal obstruction (SNOT-22) and objective rhinomanometry parameters. This makes the results more accessible and transparent.

Comment 7:

Line 254: 'significantly lower mean' – than what?

  • Response:

This has been clarified. “The Sniffin' Test Total Score in the study group had a mean of 30.235 (SD = 4.3797), with individual scores ranging from 14.75 to 38.00, indicating a relatively broad distribution of olfactory function within the group. In this group, the average nasal obstruction score in SNOT-22 had a value of 2.480 (SD = 1.1648), suggesting that while participants experienced varying degrees of nasal obstruction, the overall severity was moderate. “

  • Comment 8:

Be consistent with significant digits.

  • Response:

We have standardized the number of significant digits across tables and text for clarity and consistency.

  • Comment 9:

The title is too long and doesn't reference the main aim.

  • Response:

We have revised the title to: Impact of Pregnancy Rhinitis on Olfactory Sensitivity: A Controlled Comparative Study.

  • Comment 10:

Add citations to paragraphs 2 and 3 of the Introduction.

  • Response:

Additional references have been added to support statements in these paragraphs (marked in green).

  • Comment 11:

Clarify the description of the Sniffin’ Sticks identification test.

  • Response:

We have expanded the description of the odor identification subtest and explained the scoring system and its interpretation in the Method section (text marked in green).

  • Comment 12:

Why include 'supersmeller' in Table 2 if no participants fell into that category?

  • Response:

We have removed the 'supersmeller' category from Table 2 for clarity and relevance.

  • Comment 13:

'Functional' is misspelled.

  • Response:

Corrected. Thank you.

  • Comment 14:

Typo on line 187: 'Additionall'.

  • Response:

Corrected.

  • Comment 15:

Reorganize the Discussion. Paragraphs 3, 5, and 6 are speculative.

  • Response:

We thank the Reviewer for this constructive feedback. The Discussion has been revised (pages 270-333) to reduce speculative language and focus on evidence-based interpretation. References to potential hormonal influences and maternal behaviors have been retained but presented in a more cautious, contextualized manner.

Comment 16:

Line 343: 'progressive deterioration' is unwarranted; this was not a longitudinal study.

  • Response:

We fully agree. The phrase has been replaced with “notable reduction in olfactory sensitivity” in the Conclusions (page 9, line 343) to accurately reflect our cross-sectional study design.

Comment 17:

Add more references where needed (e.g., line 265).

  • Response:

Additional citations were added in the Discussion, including references to prior human and animal studies relevant to olfactory changes in pregnancy (marked in green).

Once again, we are grateful for the Reviewer’s thoughtful comments, which helped us improve the clarity and completeness of our manuscript. All relevant revisions have been made in the text, as indicated.

Round 2

Reviewer 2 Report

Comments and Suggestions for Authors

I appreciate the authors’ attempts to revise their manuscript based on the feedback I provided. However, there are still many issues that remain, as I address below.

The issue of confabulating the terms “threshold” and “sensitivity” has not been fully addressed. It is still unclear in the Abstract, referring now to an “odor test score”, which is meaningless in this context. It would be much better if the authors simply used the terms “detection”, “threshold” and “sensitivity” appropriately throughout the manuscript.

Table 1 – “Odor Test Score” for Detection isn’t descriptive enough. OTT = Olfactory Threshold Test – word missing in the text, line 115.

Also, the method of measuring threshold is still unclear (lines 116-122). What stimuli were presented? What was the task of the participant? If the authors were establishing the lowest concentration that a participant could detect, then how was a low score reflecting low sensitivity? This must be clarified.

Not that the final line of the manuscript refers to “odor perception test score”, which, again, is not particularly meaningful.

In the original review, I wrote “Figure 1 is unnecessary. However, it does show very large variability (I’m not sure what the error bars are reflecting), and perhaps there is something to be gleaned from the participants in the study group who reported higher olfactory impairment than those who reported less impairment?”

There was no response to this, but the figure was removed. Do the authors have any comment on the variability that was noted?

In the original review, I wrote Line 182 is vague in terms of what the “statistical analysis” was.

I don’t believe the authors have responded to this comment. See line 188 of the current manuscript.

In the original review, I wrote: I don’t follow the correlational analysis between the SNOT scores and Sniffin’ Sticks. I would like to see a plot of the data reported in the paragraph that starts on line 192. It was a positive correlation? Is the p-value of 0.30 a typo? Is that the r-value? Or should it be a p value of .03? I find it very hard to follow the results of the Spearman correlation and I don’t understand why that analysis was conducted in addition to the Pearson test.

This was addressed, but now I am left wondering why one would conduct both tests? If the data violate normality, would it not make sense to simply report the Spearman correlation? Note that the Spearman correlation is reported to be significant at p<.05, but they reported the exact p-value for the Pearson correlation, which they should also do here. If p=.05, then it is technically not significant.

In the original review, I wrote: I don’t follow all of section 3.2.1 and do not see a Figure 2. What test was conducted on line 218?

Again, not all questions are answered by the authors. I believe the analysis referred to on line 229 is not identified.

In the original review, I wrote: Table 3 is very confusing. I don’t understand the text that describes it. Where do the numbers in the text come from as they are not in the figure? And to which comparisons do the p-values in  the table the refer?

The authors’ reply was: We appreciate this comment. To improve clarity, we revised Section 3.2.2 (Results, page 4, lines 228-236) and streamlined the presentation of rhinomanometry results. The original Table 3 was removed, and the relevant data were integrated into the text with clearer descriptions. This revision avoids confusion and improves readability.

However, the previous Table 3 appears to now be Table 4 (although the text refers to Figure 3) and the analyses are still unclear. What statistical tests were conducted for the effect of pressure? What does “comparative analysis” on line 240 refer to?

Note: the formatting of Table 3 is hard to read and there are typos. The figure caption might include what “M” and “Me” refer to.

In the original review, I wrote: Section 3.2.3 should include a table or a figure.

The authors have included a table (referred to as Table 3). Seeing the current table, I understand why it would not have been included in the original paper. I don’t think this table adds much. Perhaps a figure would provide better information.

In the original review, I wrote: Section 3.3 – line 254 “significantly lower mean” – than what? Why was this analysis only done for pregnant women and not controls?

I still don’t understand the comparison being made. It seems to be a comparison between “apples and oranges” as we say.

In the original review, I wrote: Smaller statistical point – be consistent with significant digits – some are as large as 4 significant digits, which seems excessive.

The journal editors or type-setters should decide what is the appropriate number of significant digits. Note that 3 and 4 significant digits for mean Sniffin’ Stick tests, for example, seems excessive and not meaningful.

These are the other comments I made, to which the authors did not respond and it leaves me digging through where and how the text was modified:

Both paragraphs 2 and 3 of the Introduction would benefit from some additional citations. It is unclear what the citations on line 66 are really referring to. Same on line 81.

No response.

Overall, the organization of the Discussion could be improved.

First, in line with my comment on the Conclusion below, the start of the Discussion could be clearer in terms of summarizing the main findings. Is the phrase “closely associated” on line 167 warranted?

Apologies – this should have been line 267, not 167, now line 283.

Also, I would add citations on line 265 in the Discussion.

I see citations have been added, but I’m not sure that those cited specifically address this or provide any evidence for it.

Paragraph 3 in the Discussion (starts on line 273) is conjecture and no relevant data are presented in this paper; it could be moved later in the Discussion. Same is true for paragraphs 5 and 6. Should reference 13 really be included in the citations on line 272?

No response.

Paragraph 4 is more germane to the current study. It is worth noting that the same conclusion can be drawn more generally (not just for pregnant women); self-reported smell function is notoriously poorly correlated with objective measures of sense of smell. It is interesting that self-report of nasal obstruction IS correlated with the rhinomenometry measures – a point that may be worth highlighting.

No response.

Is 15 really the correct citation for line 292? It is about animals, not humans.

No response.

I can’t find citation 17. Do the authors mean this one?

Oettl, L. L., & Kelsch, W. (2017). Oxytocin and olfaction. Behavioral pharmacology of neuropeptides: oxytocin, 55-75.

No response.

Paragraph 7 seems particularly relevant and could be moved earlier and be expanded with additional references.

No response.

The Conclusions do not clearly summarize the data. The text does not really address the main aim of the study, as cited in the Abstract. Moreover, the use of the phrase “progressive deterioration in odor perception thresholds” on line 343 is unwarranted given that no longitudinal measures were taken. Again, the issue of threshold/sensitivity needs to be clarified (see above).

No response

A reference for the authors to consider including, if they are unaware of it:

Philpott, Carl M., et al. "Nasal physiological changes during pregnancy." Clinical Otolaryngology & Allied Sciences 29.4 (2004): 343-351.

This one is not a big deal, but it was also ignored.

Author Response

  1. The issue of confabulating the terms “threshold” and “sensitivity” has not been fully addressed… Abstract refers to an “odor test score”, which is meaningless. Use “detection”, “threshold”, “sensitivity” appropriately.

We thank the Reviewer for emphasizing this again. We carefully revised the terminology throughout the manuscript. “Odor test score” was removed and replaced with precise terms: “odor threshold test (OTT)”, “odor discrimination test (ODT)”, and “odor identification test (OIT)”. We clarified that a higher threshold score reflects better sensitivity (lower detection threshold).

lines 25,27,28, 117, 170,  

  1. Table 1 - “Odor Test Score” for Detection isn’t descriptive enough. OTT - Olfactory Threshold Test - word missing.

Corrected: the term now appears as “Odor Threshold Test (OTT)”.

Lines 183

  1. Method of measuring threshold is unclear. What stimuli were presented? What was the task? How does scoring reflect sensitivity?

We revised the Methods to describe the Sniffin’ Sticks procedure in detail: participants were presented with decreasing concentrations of n-butanol. The lowest concentration correctly detected determined the threshold. Lower scores indicate lower sensitivity.

Lines 116-126

  1. Final line refers to “odor perception test score” - not meaningful.

This sentence was included in the previous version but is not present in the subsequent revisions.

  1. Figure 1 is unnecessary but variability was noted. Any comment?

The Figure 1 referred to in the reviewer’s comment was included only in the first version of the manuscript and illustrated the comparison of SNOT-22 results between the control and study groups lines 226-240. In accordance with the reviewer’s suggestion, it was removed. However, we addressed and commented in variability in the Results by reporting range and SD of Sniffin’ Sticks scores in the study group and emphasizing inter-individual differences lines 280-289. The current Figure 1 refers to the comparison of the prevalence of self-reported nasal obstruction in the control group and the pregnant group.

  1. In the original review, I wrote Line 182 is vague in terms of what the “statistical analysis” was. I don’t believe the authors have responded to this comment. See line 188 of the current manuscript. In the original review, I wrote: I don’t follow the correlational analysis between the SNOT scores and Sniffin’ Sticks. I would like to see a plot of the data reported in the paragraph that starts on line 192. It was a positive correlation? Is the p-value of 0.30 a typo? Is that the r-value? Or should it be a p value of .03? I find it very hard to follow the results of the Spearman correlation and I don’t understand why that analysis was conducted in addition to the Pearson test.. This was addressed, but now I am left wondering why one would conduct both tests? If the data violate normality, would it not make sense to simply report the Spearman correlation? Note that the Spearman correlation is reported to be significant at p<.05, but they reported the exact p-value for the Pearson correlation, which they should also do here. If p=.05, then it is technically not significant.

In the revised version, we have substantially expanded the description of the statistical analysis to clearly indicate which tests were performed and under which conditions (Shapiro-Wilk for normality, Mann-Whitney U for group comparisons, Fisher’s exact for categorical variables, Pearson vs. Spearman correlations depending on distribution).

Regarding the correlation analysis, we clarified that Pearson’s correlation was used to test for a linear relationship (r=0.38, p=0.030), while Spearman’s rank correlation was performed to confirm robustness given the borderline normality of the data (rho = 0.36, p = 0.047). We agree that reporting both provides transparency, but emphasize that both analyses demonstrated a consistent significant positive association.

Section 3.1.4, lines 203-219

  1. On the original review, I wrote: I don’t follow all of section 3.2.1 and do not see a Figure 2. What test was conducted on line 218? Again, not all questions are answered by the authors. I believe the analysis referred to on line 229 is not identified.

We thank the Reviewer for pointing out this lack of clarity. In the revised manuscript, we explicitly state which tests were performed. A comparative analysis of SNOTT - 22 was assessed using Fisher’s exact test.

Figure 2 is located in line 275.

  1. In the original review, I wrote: Table 3 is very confusing. I don’t understand the text that describes it. Where do the numbers in the text come from as they are not in the figure? And to which comparisons do the p-values in the table the refer? The authors’ reply was: We appreciate this comment. To improve clarity, we revised Section 3.2.2 (Results, page 4, lines 228-236) and streamlined the presentation of rhinomanometry results. The original Table 3 was removed, and the relevant data were integrated into the text with clearer descriptions. This revision avoids confusion and improves readability. However, the previous Table 3 appears to now be Table 4 (although the text refers to Figure 3) and the analyses are still unclear. What statistical tests were conducted for the effect of pressure? What does “comparative analysis” on line 240 refer to? Note: the formatting of Table 3 is hard to read and there are typos. The figure caption might include what “M” and “Me” refer to.

\We agree that the previous Table 3 did not add substantial information in the main text. In accordance with the Reviewer’s comment, we have replaced the table with a figure that provides a clearer graphical representation of the results. Specifically, we now present a Spearman correlation between nasal obstruction in SNOT-22 and rhinometry parameters for both inhalation and exhalation at pressures of 75, 150, and 300 Pa, separately for the left and right nostrils.(Figure 2).

  1. In the original review, I wrote: Section 3.2.3 should include a table or a figure. The authors have included a table (referred to as Table 3). Seeing the current table, I understand why it would not have been included in the original paper. I don’t think this table adds much. Perhaps a figure would provide better information.

As we mentioned above we changed the table 3 for Figure 2

  1. In the original review, I wrote: Section 3.3 – line 254 “significantly lower mean” – than what? Why was this analysis only done for pregnant women and not controls? I still don’t understand the comparison being made. It seems to be a comparison between “apples and oranges” as we say.

We thank the Reviewer for pointing out this ambiguity. We have revised the text in Section 3.3 to clarify that the “significantly lower mean” refers to the OTT in pregnant women compared with the control group.

  1. In the original review, I wrote: Smaller statistical point – be consistent with significant digits – some are as large as 4 significant digits, which seems excessive. The journal editors or type-setters should decide what is the appropriate number of significant digits. Note that 3 and 4 significant digits for mean Sniffin’ Stick tests, for example, seems excessive and not meaningful.

We thank the Reviewer for this valuable remark. We agree that the initial reporting of mean values and test statistics with up to 3–4 significant digits was excessive. In the revised version of the manuscript, we have standardized the reporting of all descriptive and inferential statistics to a consistent number of significant digits (generally two for mean ± SD and effect size measures, and one or two for test statistics, depending on context).

  1. Both paragraphs 2 and 3 of the Introduction would benefit from some additional citations. It is unclear what the citations on line 66 are really referring to. Same on line 81. No response

We thank the Reviewer for this remark. We have revised paragraphs 2 and 3 of the Introduction and added additional citations to clarify and strengthen the background.

Lines 59-82, all numbers of new citations in the manuscript are marked in green

  1. Overall, the organization of the Discussion could be improved. First, in line with my comment on the Conclusion below, the start of the Discussion could be clearer in terms of summarizing the main findings. Is the phrase “closely associated” on line 167 warranted? Apologies – this should have been line 267, not 167, now line 283. Also, I would add citations on line 265 in the Discussion. I see citations have been added, but I’m not sure that those cited specifically address this or provide any evidence for it.

We thank the Reviewer for these constructive comments. We have revised the beginning of the Discussion to provide a clearer summary of the main findings before moving to interpretation. In addition, we modified the wording “closely associated” to “significantly associated,” which more accurately reflects the results. Furthermore, we have updated the citations on line to include studies that more directly address olfactory changes in pregnancy and their relationship to nasal obstruction.

Lines 296, 297,299

  1. Paragraph 3 in the Discussion (starts on line 273) is conjecture and no relevant data are presented in this paper; it could be moved later in the Discussion. Same is true for paragraphs 5 and 6. Should reference 13 really be included in the citations on line 272? No response.

 In the revised version, paragraph 3 as well as paragraphs 5 and 6 have been moved to a later section of the Discussion, where they are presented as potential implications and directions for future research. In addition, we have reviewed the citations and removed reference 13, which was not directly relevant. We have replaced it with more appropriate references addressing olfactory changes in pregnancy.

Lines 310 - 338

  1. Paragraph 4 is more germane to the current study. It is worth noting that the same conclusion can be drawn more generally (not just for pregnant women); self-reported smell function is notoriously poorly correlated with objective measures of sense of smell. It is interesting that self-report of nasal obstruction IS correlated with the rhinomenometry measures – a point that may be worth highlighting. No response.

We have revised paragraph 4 of the Discussion line to emphasize that the discrepancy between self-reported and objective olfactory function is a well-known phenomenon beyond pregnancy. We also highlighted the contrasting finding that subjective nasal obstruction did correlate with rhinomanometry measures, which further supports the internal validity of our data. Relevant citations have been added to strengthen this point.

Lines 300-309

  1. I can’t find citation 17. Do the authors mean this one? Oettl, L. L., & Kelsch, W. (2017). Oxytocin and olfaction. Behavioral pharmacology of neuropeptides: oxytocin, 55-75. No response.

We confirm that reference 17 was incorrectly formatted in the original submission. The intended citation was indeed: Oettl, L. L., & Kelsch, W. (2017). Oxytocin and olfaction. In Behavioral pharmacology of neuropeptides: oxytocin (pp. 55–75). Springer.

Lines 309, 331, 333,

  1. The Conclusions do not clearly summarize the data. The text does not really address the main aim of the study, as cited in the Abstract. Moreover, the use of the phrase “progressive deterioration in odor perception thresholds” on line 343 is unwarranted given that no longitudinal measures were taken. Again, the issue of threshold/sensitivity needs to be clarified (see above). No response

We have revised the Conclusions to provide a clearer summary of our findings and to directly address the main aim of the study. We also corrected the wording to avoid any suggestion of longitudinal inference and replaced “progressive deterioration in odor perception thresholds” with a more accurate description of significantly lower odor threshold scores in the pregnant group compared with controls. In addition, we clarified the terminology regarding threshold versus sensitivity to ensure consistency throughout the manuscript.

Lines : 362-376

  1. A reference for the authors to consider including, if they are unaware of it: Philpott, Carl M., et al. "Nasal physiological changes during pregnancy." Clinical Otolaryngology & Allied Sciences 29.4 (2004): 343-351.

We have added the reference by Philpott et al. (2004) on nasal physiological changes during pregnancy to the Introduction/Discussion, as it directly supports the background and interpretation of our findings. Lines 296, 72

Round 3

Reviewer 2 Report

Comments and Suggestions for Authors

First, let me thank the authors for responding to each point of the review, one by one, making the re-review much easier this time around.

I believe the paper has become much clearer, but I am afraid I still have a few questions/suggestions to improve clarity.

Abstract

The authors write: “Participants in the study group demonstrated a lower odor test score, indicating worse olfactory acuity and poorer identification abilities, with these effects being strong. On the other hand, the control group showed a higher level of discrimination test score compared to the study group, though this effect was weak.”

Couldn’t this be more simply written?

Here are two possibilities:

The study group demonstrated lower odor average test scores, indicating worse olfactory acuity and poorer identification abilities, with these effects being strong. In addition, the study group showed a lower discrimination test score compared to the study group, though this effect was weak.

Pregnant women scored lower on the olfactory sensitivity and identification tests, with these effects being strong. They also scored lower on the odor discrimination test, but this effect was weak.

Methods

The description of the threshold testing is much clearer. However, I think it is still unclear what the score actually reflects. What is the “mean of the last four staircase reversals”? Mean of what? I’m having trouble reconciling high scores (which seem like they should reflect higher concentrations) with the idea that high scores reflect low thresholds/high sensitivity.

Results

The description of the correlation analysis is much clearer now. The authors highlight that low SNOT-22 scores are reported for people who are score relatively low on the objective test. It might be worth mentioning the other extreme, which is also puzzling - higher SNOT-22 scores (more reported dysfunction) is correlated with higher overall Sniffin’ Stick scores? I find this the more perplexing part of the correlation.

I don’t think Figure 1 is necessary, but it is the author’s/editor’s decision. It doesn’t need to be in color.

I don’t understand lines 286-290. I thought all women were in their third trimester? Was there are result to be included here?

Figure 2 is better than the previous table, though (FYI) when printed in black and white it is very difficult to see the differences in grayscale.

Discussion

Paragraph 3 seems out of place as it breaks up the flow of the discussion of results from the current study.

The segue/transition on line 329 is under. “This suggests….” To what is “this” referring?

Author Response

We would like to sincerely thank the Reviewer for the careful evaluation of our manuscript and the constructive comments provided. We highly appreciate the time and effort dedicated to improving the quality and clarity of our work.

Comment 1 :

Abstract

The authors write: “Participants in the study group demonstrated a lower odor test score, indicating worse olfactory acuity and poorer identification abilities, with these effects being strong. On the other hand, the control group showed a higher level of discrimination test score compared to the study group, though this effect was weak.”

Couldn’t this be more simply written?

Here are two possibilities:

The study group demonstrated lower odor average test scores, indicating worse olfactory acuity and poorer identification abilities, with these effects being strong. In addition, the study group showed a lower discrimination test score compared to the study group, though this effect was weak.

Pregnant women scored lower on the olfactory sensitivity and identification tests, with these effects being strong. They also scored lower on the odor discrimination test, but this effect was weak.

Response:

We have revised the text and we adopted the first proposed version (marked green) lines: 26-30

Comment 2

Methods

The description of the threshold testing is much clearer. However, I think it is still unclear what the score actually reflects. What is the “mean of the last four staircase reversals”? Mean of what? I’m having trouble reconciling high scores (which seem like they should reflect higher concentrations) with the idea that high scores reflect low thresholds/high sensitivity.

Response

Thank you for raising this point. The threshold score in the Sniffin’ Sticks test is calculated according to the standardized protocol of the test. We followed this established standard in our work. For a detailed description of the test procedure, we added a reference to the manuscript line 116

Hummel, T., Sekinger, B., Wolf, S. R., Pauli, E., & Kobal, G. (1997). ‘Sniffin’ Sticks’: Olfactory performance assessed by the combined testing of odor identification, odor discrimination and olfactory threshold. Chemical Senses, 22(1), 39–52. https://doi.org/10.1093/chemse/22.1.39

In the Sniffin’ Sticks threshold test, n-butanol is presented in 16 serial dilutions (dilution step 1 corresponds to the highest concentration, and step 16 to the lowest concentration). Therefore, a high score indicates that the patient perceived the odor at the most diluted solution, meaning they were highly sensitive even to the lowest concentration.

The threshold score is defined as the mean concentration level of the last four staircase reversals. This method is an integral part of the validated procedure and ensures comparability across studies. The staircase paradigm requires the concentration to be increased after an incorrect choice and decreased after a correct choice in two consecutive trials. The individual threshold score is calculated as the arithmetic mean of the dilution steps corresponding to the last four such staircase reversals.

Thus, the “mean of the last four staircase reversals” represents the average dilution step at which the participant was able to detect the odorant reliably. Because higher dilution steps correspond to lower concentrations, higher threshold scores indicate that the subject could detect the odorant at lower concentrations, reflecting greater olfactory sensitivity (i.e., a lower detection threshold). Conversely, lower scores indicate that the subject required higher concentrations to detect the odorant, reflecting reduced sensitivity. The final threshold score represented the mean of the last four staircase reversals. Higher scores indicated better olfactory sensitivity (we added this information 124-126), whereas lower scores reflected reduced sensitivity.

Comment 3
The description of the correlation analysis is much clearer now. The authors highlight that low SNOT-22 scores are reported for people who are score relatively low on the objective test. It might be worth mentioning the other extreme, which is also puzzling - higher SNOT-22 scores (more reported dysfunction) is correlated with higher overall Sniffin’ Stick scores? I find this the more perplexing part of the correlation.

We thank the reviewer for this comment and have added the following text to the manuscript:

It is worth noting that while low SNOT-22 scores generally correspond to lower objective olfactory performance, some participants may report high SNOT-22 scores and still achieve high Sniffin’ Sticks results. This could be due to factors such as nasal obstruction, sleep disturbances, or general discomfort affecting their self-reported sinonasal burden, even when their objective olfactory function remains intact. This highlights the importance of considering both subjective and objective measures when evaluating olfactory performance.

We added this information to the line: 333-339

I don’t think Figure 1 is necessary, but it is the author’s/editor’s decision. It doesn’t need to be in color.

We removed figure 1.

I don’t understand lines 286-290. I thought all women were in their third trimester? Was there are result to be included here?

To clarify, all participants in the study group Line: 94-99 were indeed in their third trimester of pregnancy. The text in lines 286–290 referred only to the gestational age range of our participants (28th to 41st week) and was not intended to suggest comparisons across different trimesters.

Figure 2 is better than the previous table, though (FYI) when printed in black and white it is very difficult to see the differences in grayscale.

We revised Figure 2 by replacing the grayscale shading with a palette of contrasting colors. In addition, all correlation values are now directly annotated within the cells, ensuring that the figure remains interpretable even when printed in black and white.

Comment 3

Paragraph 3 seems out of place as it breaks up the flow of the discussion of results from the current study.

We removed paragraph 3

The segue/transition on line 329 is under. “This suggests….” To what is “this” referring?

To avoid ambiguity, we have revised the sentence to explicitly state the reference. We added line 320-321 marked green